# The Mechanism of Metal Homeostasis in Plants: A New View on the Synergistic Regulation Pathway of Membrane Proteins, Lipids and Metal Ions

**DOI:** 10.3390/membranes11120984

**Published:** 2021-12-15

**Authors:** Danxia Wu, Muhammad Saleem, Tengbing He, Guandi He

**Affiliations:** 1College of Agricultural, Guizhou University, Guiyang 550025, China; gs.wudx19@gzu.edu.cn; 2Department of Biological Sciences, Alabama State University, Montgomery, AL 36104, USA; msaleem@alasu.edu; 3Institute of New Rural Development, West Campus, Guizhou University, Guiyang 550025, China

**Keywords:** specific transmembrane proteins, protein-lipid interactions, plant metal homeostasis

## Abstract

Heavy metal stress (HMS) is one of the most destructive abiotic stresses which seriously affects the growth and development of plants. Recent studies have shown significant progress in understanding the molecular mechanisms underlying plant tolerance to HMS. In general, three core signals are involved in plants’ responses to HMS; these are mitogen-activated protein kinase (MAPK), calcium, and hormonal (abscisic acid) signals. In addition to these signal components, other regulatory factors, such as microRNAs and membrane proteins, also play an important role in regulating HMS responses in plants. Membrane proteins interact with the highly complex and heterogeneous lipids in the plant cell environment. The function of membrane proteins is affected by the interactions between lipids and lipid-membrane proteins. Our review findings also indicate the possibility of membrane protein-lipid-metal ion interactions in regulating metal homeostasis in plant cells. In this review, we investigated the role of membrane proteins with specific substrate recognition in regulating cell metal homeostasis. The understanding of the possible interaction networks and upstream and downstream pathways is developed. In addition, possible interactions between membrane proteins, metal ions, and lipids are discussed to provide new ideas for studying metal homeostasis in plant cells.

## 1. Introduction

Heavy metal stress (HMS) negatively affects plant growth and reproduction, and thus, it can cause the loss of essential agronomic and other agroecological traits in plants [1]. Consequently, the yield and quality of crops are seriously affected, while food with excessive accumulation of heavy metals (HMs) is one of the major threats to human health and natural ecosystems [2]. Plants have specific mechanisms to cope with HMS, and they have evolved a series of strategies such as ion sensing, signal transduction, and segregation detoxification to ensure their optimum survival and reproduction under a certain level of HMS [3,4,5]. Recently, some advances have been made in the understanding of plant stress responses, especially downstream pathways, with the discovery of several functional membrane protein gene clusters that play important roles in regulating intracellular metal homeostasis [6,7,8,9]. However, these complex network cascades need to be studied with respect to the metal homeostasis regulation mechanisms in plants.

Recent studies have shown that the plant growth stage, from seedling to maturity, determines the formation of metal homeostasis mechanisms and tolerance in plants. Of course, the HM contents in fruits during the reproductive maturity stage of plants are significantly influenced by the metal homeostasis, though the systematic accumulation pathways and functional cascade relationships of HMs across different plant growth stages are still unclear [10,11]. Therefore, it is of great significance to explore the homeostasis mechanisms of HMs in plant cells. Currently, some studies have demonstrated that plants can regulate the balance of metal ions in the cell by increasing the rate of synthesis of various proteins and/or compounds [12,13,14,15]. This involves the molecular response signals of plants to HMS along with the regulation and transcription of some membrane proteins. In the plant cell, these metals cause the production of reactive oxygen species (ROS), which trigger the activation of various signal transduction pathways [16]. The key signal components involved in HMS are mitogen-activated protein kinase (MAPK), calcium, and hormonal signals. The MAPK signal has some important components and types, such as the MAP kinase kinases (MAPKKs) and mitogen-activated protein kinase kinases (MAPKS) [17,18]. The calcium signaling pathway uses a variety of calcium-sensitive proteins for Ca^2+^, such as calmodulins (CaMs), calmodulin-like proteins (CMLs), calcineurin-like proteins (CBLs), and calcium-dependent protein kinases (CDPKs). They activate and route different signal pathways along the circuit [19]. Regarding the role of hormone signal transduction, abscisic acid (ABA) [20], jasmonic acid (JA) [21], and citric acid (CA) [22] play important roles in metal tolerance in plants. In addition to the signal components, other regulatory factors, such as miRNAs and different types of transmembrane transporters, also play important roles in regulating HMS in plants [23,24]. Specific membrane proteins can bind different types of metal ions as substrates [25,26], thus making them capable of ion transport across the cell membrane, which is regulated by the interactions between membrane proteins, ions, and lipid types and contents. Some lipids bind to specific sites on membrane channels, and thus regulate the transport of metal ions inside and outside of the plant cell.

This paper reviews the key role of membrane proteins in the simultaneous regulation of other signal elements (MAPK and ABA signals) and regulatory factors (transcription factors, microRNAs) in metal tolerance mechanisms in plants. Moreover, we also emphasize the role of membrane protein-lipid-ion interactions in metal homeostasis. We anticipate that this review will provide new insights into the mechanisms of metal homeostasis in plant cells.

## 2. Role Cell Membrane Proteins in HMS Tolerance in Plants

### 2.1. The Reorganization of Endomembranes

When plants are exposed to HMS, the plasma membrane, as the first biological barrier, plays an important role in protecting cells from the damage and toxicity caused by the metals [27]. Plasma membranes are mainly composed of membrane proteins and lipids [28]. Here, we will discuss the role of proteins in membrane remodeling. A large number of studies have shown that the contents of lipids, phospholipids, glucolipids, and sterols in the membrane are significantly changed under abiotic stresses such as HMS and high salinity [29,30]. This means that in response to abiotic stresses, plants reshape their membrane lipid composition and use this process to recruit many membrane proteins to resist HMS. For example, the metal ion efflux pump [31], glutathione transferase [32], aquaporin [33] and some other proteins have been related to toxic metal ion efflux, compartmentalization, and detoxification. Therefore, the reorganization of plant endomembranes under HMS is an important process to ensure lipid redistribution and metal homeostasis, because these membranes determine the location and functions of membrane proteins.

### 2.2. Detoxification Mechanism

Plants have evolved various mechanisms to detoxify excessive or small amounts of metal-like or heavy metal (HM) ions that include, but are not limited to, the following: (1) reducing the absorption of HMs, (2) compartmentalization of metals, and (3) reducing the toxicity of metal ions (via chelation) [34,35]. These mechanisms are closely related to ion channel proteins and transporters in the membranes [24]. The members of these proteins are diverse and have different specificity for metal ions, while they differentially regulate the transport of metal ions in the horizontal or vertical directions in plant cells (Figure 1). In plants, the main metal membrane protein families include Zrt/Irt-like proteins (ZIPs) [36], ATP-binding cassette (ABC) transporters [37], cation diffusion facilitator (CDF) transporters [38], metal tolerance proteins (MTPs) [39], natural resistance-associated macrophage proteins (NRAMPs) [40], cation/proton exchanger (CAXs) transporters [41], aquaporins (AQPs) [42] and many others. The members of the ZIP family are mainly regulated by Zn and Fe, and they also participate in the transport of Zn, Fe, Mn, Cd, and Co in the roots of plants. OsZIP9 promotes the absorption of Zn and Co by rice roots [43]. VsRIT1 is involved in Cd uptake in alfalfa roots [44]. The ABC transporters belong to the large-scale protein family and are widely distributed in a variety of organisms. ABC transporters can be divided into eight ABCA-ABCI subfamilies in terms of their phylogeny. The ABCH group is not reported in plants [45]. Most of the members of the ABC family are located in the plasma membrane and tonoplast, while tonoplast-located ABCs play the role of metal detoxification by immobilizing metal ions in vacuoles [46]. The CDF transporters play important roles in the homeostasis and tolerance of bivalent metal ions in plants. It can isolate metal ions in vacuoles and consequently reduce the toxicity of HMs in plants [47]. The NRAMPs are an evolutionarily conserved important metal transporter family and are responsible for the absorption and transport of divalent cations [48]. Studies have shown that vacuolar localization AtNRAMPs 3/4 are involved in the regulation of manganese homeostasis in *Arabidopsis thaliana* [49], while AtNRAMP1 is known to reduce the toxicity of Mn and Zn in peanut plants [50].

There are differences in the absorption and transportation mechanisms of HMs between leaves and roots (Figure 2). Roots absorb HMs from the soil, while membrane proteins store them in vacuoles and/or transport them to the xylem. Then, leaves absorb HMs via xylem and can transport them to different plant organs. Metal ions are easily bound to the carboxyls of polysaccharides and uronic acid. These are passively transported to the root cell surface, and then they permeate to the roots via diffusion. Metal ions enter the xylem through the symplast, apoplast, and apoplast coupling pathways. Meanwhile, both root pressure and transpiration also regulate the transfer of free and chelated HMs from the roots to aboveground plant tissues [51,52]. This process is also influenced by several important membrane transporters [53]. The transport process of HMs by leaves is similar to that of photosynthesis products. In leaves, the HM ions are desorbed in the apoplast and are then combined with lower mesophyll cells. Subsequently, these metal ions and their complexes are transported from leaves to other plant organs through the phloem [54,55]. However, there is no clear evidence to confirm the role of specific chelating agents and transporter genes in metal transport after absorption in the plant leaves. The mechanisms of compartmentalization and transport of HMs and key physiological processes in leaves need to be further studied.

### 2.3. Important Aquaporin Family

Aquaporins (AQPs) are divided into 5 subfamilies based on function: plasma membrane intrinsic proteins (PIPs), tonoplast intrinsic proteins (TIPs), nodulin 26-like intrinsic proteins (NIPs), small basic intrinsic proteins (SIPs), and uncategorized intrinsic proteins (XIPs) [56]. The AQPs play an important role in the regulation of plasma membrane permeability, which is mainly involved in maintaining cell homeostasis [57,58]. The study of its regulation mechanism is one of the hot topics in the field of plant sciences, especially with respect to abiotic stress tolerance in plants [59,60,61]. The AQPs have been heavily investigated over the past 20 years. The transport activity, substrate specificity, and role of AQPs in HMS have been widely reported [62,63]. Under HMS, the chemical composition and physical properties of the lipid bilayer are changed. Additionally, the stability of lipid around aquaporin is affected, thus regulating the activity of aquaporin [64,65]. However, there are a large number of AQPs in plants, and their transport substrates, structural characteristics, and signal regulation forms are different [66,67]. The specific roles of different subgroups in HMS have not been revealed. The molecular pathways of AQP under HMS are summarized in Figure 3. Under HMS, the ABA signal (mainly) is activated, resulting in a series of downstream stress responses, thus regulating AQPs at the transcriptional level [68]. Translated AQPs are phosphorylated and heteromerized, and they actively participate in the regulation of cell pH, Ca^2+^, and other metal ion homeostasis. Members of different AQP subfamilies (SIPs, PIPs, TIPs, NIPs, XIPs) function in different subcellular locations to maintain the health and stability of the cell environment. Through strict transcription, translation, subcellular transport, and gated regulation, they form a fine regulatory network to limit the inflow of metal ions and maintain normal cellular metabolism [69]. In addition, ABA induces tryptophan-rich transporters and E3 ubiquitin ligases to mediate the low expression of some aquaporins through autophagy and lysosomal degradation pathways to limit the transport of metal ions [70]. In summary, AQPs play important roles in maintaining ion homeostasis and promoting plant growth under normal and stressful conditions. Different AQPs respond to stress by changing their structure and activity, which makes a significant contribution to the physiological processes of stressed plants. However, the regulatory patterns expressed in the complex network between them and the underlying signaling pathways are still unclear.

## 3. The Role of Lipid Regulation in Plant Responses to HMS

### 3.1. Cascade Signal Transduction

Membrane proteins play very important roles in many physiological processes of organisms, such as cell proliferation and differentiation, energy conversion, signal transduction, material transport, etc. [76,77,78,79]. They are mainly divided into transporters, ion channels, and receptors. As one of the serious abiotic stressors, the bioaccumulation and bioamplification of HMs in the environment pose a great threat to the plant growth and development [80]. HMs, at toxic levels, can interact with several important cellular biomolecules, such as nuclear proteins and DNA, to induce excessive increases in the synthesis of ROS, thus resulting in serious morphological, physiological, and biochemical abnormalities in plants [81]. Plants have evolved complex signaling mechanisms to relieve the toxicity caused by HMs [82]. These mechanisms are accomplished by a cascade of three steps: reception of stimuli, transduction of intracellular and extracellular signals, and enzymatic or non-enzymatic reactions (Figure 4).

### 3.2. Abscisic Acid

ABA is an important plant hormone which plays an important role in many stages of plant growth and development [83]. Its biosynthesis, signal transduction, and downstream reactions have been extensively studied [84,85]. Exogenous applications of ABA are known to relieve the toxic effects of HMs and metal-like substances on plants [86]. For example, exogenous ABA can improve plant tolerance under Cd and Pb stresses to some extent [87,88]. ABA alters the movement of Cd and Pb from underground to aboveground plant tissues in a variety of ways [89]. First of all, ABA is capable of promoting the alteration of apoplastic root barriers to inhibit the radial transport of Cd to the stele [90]. Second, ABA inhibits Cd and Pb transport from roots to aboveground tissues by regulating stomatal closure [91]. Third, ABA alters the redistribution of HMs in plants by activating the expression of membrane transporters [92,93]. Finally, ABA may also inhibit the movement of nitrate from roots to the ground mediated by NRT1.5 through enhancing the activities of vacuolar-type H^+^-ATPase (V-ATPase) and vacuolar-type H^+^-PPase (V-PPase), thus changing the distribution of Cd in roots and aboveground tissues [94].

Many membrane proteins involved in Cd and Pb transport are identified in crop and model plants such as rice and arabidopsis, which are directly or indirectly regulated by the ABA. For example, under Cd stress, the plasma membrane-localized HMAs, such as OsHMA2, AtHMA2, and AtHMA4, relocate Cd into the stele, and then move it to the shoot tissues [95,96,97]. However, the tonoplast-localized AtHMA3 and OsHMA3 immobilize Cd in the root vacuoles, thus limiting its transport to stems, leaves, and grains [98,99]. Natural variation of the HMA3 gene is a key determinant of the transfer and accumulation of Cd to the upper plant parts in arabidopsis [100], rice [101], and soybeans [102]. In addition, some members of the NRAMPs family have been reported to transport free Cd ions. For instance, NRAMPs 1/3/4 are up-regulated in arabidopsis under inoculation with ABA-catabolizing bacteria [92]. The ABA-induced transcription of SaHMAs 2/3/4 to promote the resistance and root-to-stem migration of Cd is reported in the Cd-hyperaccumulating ecotype (HE) of *Sedum alfredii* [93]. In addition, some membrane proteins are indirectly involved in ABA-related Cd tolerance. In arabidopsis, the nitrate membrane protein NRT1.5 relocates nitrate from roots to shoot tissues (buds) [103], while NRT1.8 enhances Cd tolerance by eliminating nitrate from the xylem vessels [104]. Exogenous ABA may inhibit the expression of NRT1.5 but not that of NRT1.8, which increases the accumulation of nitrate in roots and ultimately enhances the resistance of plants to Cd stress [94].

### 3.3. Plant TFs and miRNAs Play Important Roles in Regulating Metal Homeostasis

In addition to the signal components, other regulatory factors such as transcription factors and microRNA also play an important role in regulating HMS. Transcription factors are proteins that can bind to specific sequences upstream of gene sequences. They regulate biological processes by changing gene expression and play an important role in plant growth and development under stress conditions [105]. Studies have shown that several transcription factors, such as WRKY [106], ERF [107], MYB [108], bZIP [109], NAC [110], bHLH [111], and others, respond to HMS in plants. The WRKY transcription factor family is one of the largest in plants; it is widely involved in plant growth and development, and it plays a very important role in improving plant tolerance under abiotic stresses such as HMS, drought, salinity, freezing injuries, etc. [112]. GmWRKY142 directly regulates the stress-related genes to reduce the Cd absorption and actively regulate Cd tolerance in plants (e.g., *Glycine max*) [113]. The transcription factor WRKY13 positively regulates Cd tolerance in arabidopsis by activating the PDR8 (ABC transporter), a Cd effector pump that reduces Cd accumulation by pumping Cd^2+^ from the cytoplasm to the ectoplasm [114]. In addition, in arabidopsis, three bHLH transcription factors, FIT, AtbHLH38 and AtbHLH39, are involved in its response to Cd stress [115]. Double overexpression of FIT/AtbHLH38 and FIT/AtbHLH39 showed that transgenic plants were more tolerant to Cd stress than wild type plants. Double overexpression of FIT/AtbHLH38 and FIT/AtbHLH39 improved the tolerance of Cd, mainly due to the interaction between FIT and AtbHLH38 or AtbHLH39, by constitutively initiating the expression of some genes (such as HMA3, MTP3, IREG2 and IRT2) [115]. Overall, these relocated a large amount of absorbed Cd in the roots and reduced its transport to the shoot. At the same time, the interaction between FIT and AtbHLH38 or AtbHLH39 constitutively promoted the expression of nicotianamine (NA) synthase genes (NAS1 and NAS2), which catalyzes NA synthesis in plants [116]. Because NA is the main chelate for activation and transport of iron in plants, its increase can enhance the transport of iron ions to the shoot under Cd stress, thus alleviating the complications of iron deficiency caused by Cd stress.

MicroRNAs (miRNAs) are small non-coding RNAs composed of 20–24 nucleotides which regulate gene expression by base pairing in complementary mRNA sites [117,118]. Each miRNA can have multiple target genes, and several miRNAs can also regulate the same gene. Studies have shown that miRNAs play an important role in the homeostasis mechanisms of HMs in plants [119]. There are complex gene regulatory networks in plants under HMS, and metal-regulated miRNAs and their target genes are a small part of them [23]. The network controls a variety of molecular mechanisms, including uptake, transport, chelation of HMs, protein folding, antioxidant response, and hormone signaling (Figure 5).

The uptake and transport of HMs in plants are mainly accomplished by membrane proteins. Current studies on miRNA indicate that most miRNAs are directly involved in plant tolerance to HMS through the transcription of membrane proteins. Some of these miRNAs are not only directly involved in the HMS, but they also play important roles in plant growth and development [120]. Secondly, miRNAs play an important role in metal chelation. The ATP sulfurylases (ATPS) and sulfate transporters (SULTRs) are the target genes of miRNA, and they can induce the expression of glutathione (GSH) and phytochelatin (PC) genes. The miRNAs perform a function in metal chelation by regulating ATPS and SULTRs [121]. The NRAMPs are one of the most important metal transporter families in plants. BnNRAMP1b is the target of miR167 in *Brassica napus*, and the expression framework under Cd stress indicates that miR167 regulates BnNRAMP1b post-transcriptionally. Similarly, rice NRAMP3 is the target gene of miR268, and the overexpression of miR268 inhibits the growth of seedlings under Cd stress [122]. Similarly, rice NRAMP3 is the target gene of miR268, and the overexpression of miR268 inhibits the growth of seedlings under Cd stress [123]. In addition, miRNAs are involved in plant hormone signal transduction. For example, miRNAs are abundant in rapeseed and rice, and some appear to influence genes associated with auxin signaling. For example, the transport inhibitory response 1 (TIR1) gene, as the target gene of miR393, interacts with the AUX/IAA protein as an auxin receptor [124].

## 4. Lipid Regulation in Plants under HMS

### 4.1. Membrane Fluidity

Membrane fluidity helps to maintain cell function, but it is easily affected by abiotic stressors such as HMs [125]. However, the dynamic behavior of lipids in the bilayer represents the fluidity of the membrane, which is defined by the degree of molecular movement and disorder that make up the lipids [126]. Lipid regulation is usually achieved by changing membrane permeability, membrane protein arrangement, and transmembrane transporter activity [127]. It was found that plants adapted to HMS responded to HMS by changing the unsaturated level and proportion of different lipids in the membrane (Figure 6).

### 4.2. Sphingolipids

Sphingolipids are structural components of lipid bilayers that act as signaling molecules in many cellular processes [128]. Plant sphingolipids are important factors in dealing with complex environmental stresses, including HMS [129]. In addition to being an important structural element of plant membranes, sphingolipids are also involved in a variety of cellular and regulatory processes, including growth, development, vesicular transport, and tolerance against external stresses [130,131]. Sphingolipids play an important role in HMS in plants, including in: (1) signal transduction in programmed cell death [132], (2) structural components, (3) signal messengers [133], and (4) interactions with the ABA signal pathway [134]. Although the functional and structural characteristics of different types of sphingolipids have been studied, the mechanisms of sphingomyelin in plant HMS tolerance are not clear. When plants are stressed, a large number of lipids are remodeled, and sphingolipids may be a key factor in the development of this coordinated response. Nevertheless, the mechanisms of sphingolipids and the associated expression of stress genes and their roles in HMS tolerance in plants requires further studies.

## 5. Interactions between Membrane Proteins and Lipids under HMS

All membranes, which are mainly composed of lipids and proteins, are highly dynamic and experience a large variety of interactions [135]. The complexity of these interactions can lead to the diversification of the structures and functions of the membrane proteins. Therefore, under HMS, the membrane proteins can promote intercellular communication, thus leading to the regulation of intracellular and extracellular homeostasis along with the detoxification of metal ions through energy transmission and signal transduction in plant cells. With the deepening of liposomics and structural analysis of membrane proteins, the specific sites and functions of lipid-binding membrane proteins can be visualized and studied. For example, applications of recent state-of-the-art analytical approaches such as X-ray crystallography, nuclear magnetic resonance (NMR) [136], electron paramagnetic resonance (EPR) [137], and native mass spectrometry (MS) combined with molecular dynamics (MDs) [138] have now enabled us to further study the properties of lipids that maintain the structure and functions of the membrane proteins. Thus, these approaches may provide new opportunities for an in-depth understanding of the processes underlying the interactions between lipids and membrane proteins and their role in HMS tolerance in plants.

### 5.1. Membrane Proteins and Lipid Remodeling

HMS can cause the reorganization of plant endomembranes, wherein the redistribution of lipid-membrane protein binding is closely related to the regulation of metal homeostasis to a large extent [139]. Because of the position of membrane proteins in lipid bilayers, the structures and functions of the membrane proteins may be related to their positions in the lipids [140]. Different types of membrane proteins play different roles and can regulate ion homeostasis in different ways. Interestingly, membrane proteins can bind nonspecifically to a variety of lipids, which are usually used as protein adsorption solvents (cyclic lipids), while other small amounts of lipids can specifically bind to the membrane proteins as cofactors (non-cyclic lipids) to complete the process of remodeling [141]. Specific lipids are like buttons, while membrane proteins are like clothes, and thus these lipids precisely fix membrane proteins in their correct place. Meanwhile, the solvent lipids, which are also known as cyclic lipids, can form stable annular lipid shells around the membrane proteins and can further immobilize the membrane protein [142]. Therefore, they can jointly define the position of the membrane protein and ensure the stability of the membrane skeleton. In addition, these lipids can also interact with membrane proteins through configurational changes to regulate the HMS tolerance in plants.

During endomembrane reorganization, a small number of lipid-membrane protein interactions have also been reported in microorganisms (Figure 7), and their types are related to their structures and substrates. A common feature is that their locations and functions depend on the complexity of the membrane environment. As a cofactor, lipids (acyclic lipids) usually bind between the transmembrane helices of the membrane proteins, either within the membrane proteins and/or at the multi-subunit protein interface (Figure 7A–D). However, under HMS, a large number of metal ions are free, which may lead to the substitution and/or inactivation of these sites depending on their charge intensity [143,144] and the head groups of these lipids [145]. These conditions may trigger a series of membrane protein stress responses and signals, which could lead to alterations in the transmembrane domain of the membrane proteins [146].

According to the affinity between membrane proteins and lipids, the lipids that play a key role in intimal remodeling can be divided into bulk, annular and non-annular lipids. Bulk lipids (without direct contact with membrane proteins) expand rapidly in the bilayer. At the same time, annular lipids bind to the membrane proteins through hydrophobicity to form stable complexes and manipulate the membrane proteins to function and allow material exchange, which depends on the hydrophobicity of the membrane and the charge distribution of the lipid-membrane protein binding surface. In particular, non-annular lipids have multiple functions in regulating the membrane proteins because their different configurations can bind to membrane proteins to form very stable structures. Studies have found that they remain bound together even after proteins are washed, purified, and/or crystallized. This stable structure may play an important role in HMS in plants. For example, non-annular lipids act as cofactors to block ion conduction and regulate the activity of membrane proteins.

### 5.2. Metal Coordination of Lipids with Membrane Proteins

In recent years, great progress has been made in the structural analysis of membrane proteins. However, the interactions between lipid proteins and membrane proteins in complex cellular environments remain understudied. The recent emergence of X-ray free-electron laser (XFEL) femtosecond crystallography offers an innovative opportunity to explore the elusive membrane proteins and their complexes [151]. When combined with other techniques such as electron paramagnetic resonance (EPR), molecular dynamics (MDs), atomic force microscopy (AFM) [152], and fluorescence resonance energy transfer (FRET) [153], the metal ions and lipids that interact directly with membrane proteins, as well as the interaction sites, and even the covalent bond, can also be profiled and studied. Advances in instrumentation and technology provide unprecedented opportunities to investigate the synthesis of lipid-protein complexes. At present, the crystal structures of membrane protein complexes are mostly reported with the non-cyclic lipids [138]. It has been shown that lipids can protect membrane proteins by reducing deuterium absorption, while the sites protected by the lipid-binding can be studied [154,155]. Some membrane proteins exhibit metal coordination activity only in the presence of specific types of lipids (Figure 8). These protein complexes can be purified, and their crystal structures can be visualized mechanistically. Most of these lipids are non-annular lipids. However, studies of these types of protein-lipid interactions in plants are highly limited. It is anticipated that future research can rely on the existing advanced technology and jointly use it to analyze the structure of protein complexes more widely, thus providing the possibility of understanding the action mechanisms of the membrane lipids in complex cellular environments under HMS.

### 5.3. Special Ion Channels as Lipid Sensors

The ion channel is a kind of membrane protein that promotes the passage of various chemicals through membranes. Channels open or close due to several factors, such as transmembrane potentials [156], compound stimulation [157], and membrane contraction forces. Under the influence of these factors, the channel undergoes a structural change from a closed to an open state, thus allowing the substrate to move along the electrochemical gradient. Since the selective permeation of cell membrane to sodium, potassium and chloride ions was discovered, the channels have been extensively studied. The ion channels of excitable cells respond to the synaptic stimuli and transmit action potentials through the establishment of resting membrane potentials [158]. In recent years, researchers have carried out a significant amount of work on the structures and functions of the membrane channels, particularly on using them as biosensors to monitor the development and applications of new molecules to study their role and activities in different cellular functions [159,160,161,162,163]. According to the methods of sensing external stimuli, ion channels are divided into three categories: (A) voltage-gated channels, (B) ligand-gated channels and (C) mechanosensitive channels. They respond to lipids in different ways (Figure 9).

The voltage-gated channels mediate the transmembrane movement of K^+^, Na^+^, Ca^2+^ and Cl^-^ in response to membrane potential changes [86]. This channel contains four homologous domains; each domain consists of six transmembrane helices (S1-S6). The S1-S4 is responsible for the induction of electrical signals, while the S5-S6 is responsible for the hole formation [164]. The ligand-gated channels, which are transmembrane proteins, are a common part of eukaryotic signal cascades. Plant cyclic nucleotide-gated channel (CNGC) is a cationic channel with a tetramer structure that allows the diffusion of univalent and divalent cations such as Ca^2+^ and K^+^ [165,166]. The CNGC family contributes to the absorption and transport of HM ions in plants [167]. Mechanosensitive channels, which are usually multimers, are the umbrella term for pore proteins that mainly respond to the mechanical stimuli. With the stimulation of mechanical force, the transport of substances (mainly ions) across the membrane is allowed, and the mechanical signals can be converted into electrical and/or chemical signals [168]. Mechanosensitive channels in plants may be activated by external mechanical stimuli such as touch, wind, salinity, and gravity [169,170]. Studies have shown that the homologue of bacterial mechanically-sensitive channels (MSCs) in plants can reduce the intracellular osmotic pressure by releasing small permeable substances in a process controlled by the membrane tension [169].

### 5.4. Role of Membrane Proteins in the Lipid Outflow

The plasma membrane is fundamental for cellular transport by using the lipid bilayer to form a tight barrier that allows the selective exchange of substances between the cell and external environmental factors. Membrane lipids modulate the functions of membrane transporters in two ways: one is to bind closely and specifically to the transporters, while the other is to regulate the functions of the transporters through the diversified lipid bilayer [170,171].

Members of the ABC family are involved in the transport and redistribution of various lipids and protein-lipid complexes [172]. The ABCs are mainly located in the plasma membrane, though some of them are also located in the microregions of cellular fluids (cholesterol and sphingomatidylin). These are also reported in other cellular organelles such as the Golgi apparatus and endoplasmic reticulum [173,174]. In general, apolipoprotein and albumin are receptors for most sphingolipids that are substrates for the ABC transporters in animals (Figure 10). Although studies on this aspect in plants are limited, future studies are expected to fill the gap.

### 5.5. Synergistic Participation in Specific Binding of Metal Ions

Plant membranes are mainly comprised of phospholipids and contain a bewildering array of complex components such as sphingomyelins, sterols, carbohydrates linked by glycosylation, and membrane proteins [175]. The separation of cells from their external environment cannot be maintained solely by lipid barriers. In the case of organisms and cells exposed to HMs, the membrane proteins actively adjust the different conditions on both sides of the lipid barrier [176]. Membrane proteins may cross the phospholipid bilayer and/or interact with the polar heads of lipids, thus directly and indirectly affecting the structure of cell membranes and even the composition of membranes [177,178]. Notably, membrane proteins also need to allow controlled signals to occur through the structural barriers of the cell membrane (Figure 11). The structures and functions of membrane proteins are affected by the lipids, as well as the interactions between proteins and specific lipids [179].

Plants are inevitably exposed to environmental stresses, such as metal contamination, while membrane proteins can induce mechanisms underlying plant tolerance to different stresses [180]. The plants’ interactions with HMs produce several cellular, physiological, and biochemical responses. Of course, these reactions are influenced by a complex transduction network of multiple signal components in plant cells, which eventually help plants counter HMS by synthesizing metalloproteins [181]. Usually, different proteins respond to different kinds and concentrations of metals, as described in Table 1.

## 6. Analysis of Membrane Protein-Metal Binding Domains and Interaction Based on the Novel Metallomics Database

Trace metals are inorganic elements; all organisms need them for their growth, development, and other cellular functions. Cells need metal ions as cofactors to assemble and activate metalloproteins. Most metals bind directly to specific sites on membrane proteins, while a few metals must form metal-containing cofactors or complexes to insert into the target protein [226]. Metalloproteins are types of protein with large quantities and varieties in the cellular proteome. They not only catalyze an array of important biochemical reactions but also demonstrate unique structures and regulatory functions [227]. It is speculated that about a third of proteins must bind with metals for their biological functions, and almost half of the enzymes must be combined with one or more metal ions for their functioning. The number of metalloproteins in the family depends on the number and types of metals they bind in the cell. There are estimated to be hundreds of zinc-related protein families, while the nickel-dependent type protein families number fewer than 10 [228]. The existence of metalloproteins requires strict regulation of metal metabolism and homeostasis in order to maintain the appropriate metal concentrations in the cells while avoiding toxic effects during transport, storage, detoxification, and excretion processes in cells [229].

In recent years, with the development of genome sequencing technology, the amount of genomic data has increased exponentially, so there is an urgent need to develop bioinformatics algorithms to predict new metalloproteins or even search for a whole set of metalloproteins. Many computational tools and methods have been developed to predict metalloprotein genes and metal-binding sites, especially related to Zn and Fe [181] (Table 2).

## 7. Conclusions and Prospects

In recent years, there have been some studies aimed at understanding the mechanisms of plant resistance to metal stresses in the environment. Membrane proteins are involved in metal uptake by lipid regulation and enhance the plant antioxidant reduction system to resist diffusion and metal toxicity in the cells. These cascade reactions involve complex and orderly mechanisms of intracellular and extracellular co-regulation of homeostasis, which are designed to convert extracellular stimuli into intracellular responses. For example, other regulatory factors, such as membrane proteins, transcription factors, and miRNAs, also play important roles in plants’ tolerance to HMs. In this paper, the latest studies investigating the role of membrane proteins in alleviating metal-induced stress in plants by regulating different signaling and hormonal metabolism pathways are reviewed. By doing so, we attempt to develop an improved understanding of the mechanisms underlying the membrane protein-metal ion-lipid interactions and their roles in metal homeostasis in plants.

Membrane proteins regulate stress response mechanisms by promoting the movement of various molecules and ions across cell membranes while maintaining basic cellular processes such as ion dynamics and balance, osmotic regulation, signal transduction, and detoxification. With the rapid development of genome sequencing technology, many membrane proteins have been discovered, alongside their involvement in HMS tolerance in plants, including transporters (CDFs, ZIPs, ABCs, HMAs, NRAMPs, AQPs) and three types of ion channels (voltage-gated, ligand-gated, and mechanosensitive channels). However, the structures and functions of membrane proteins are regulated by lipid bilayers and protein-lipid interactions. Some lipids act as true second messengers that bind to specific sites that control channels and gating of the membrane proteins. Other lipids play a nonspecific role by changing the physical environment of channels and membrane proteins, especially the protein-membrane interfaces. In addition, many important lipid signals occur by forming the membrane domains, which influence the functions of membrane proteins, protein-protein interactions, and the turnover of surface membranes. It is worth noting the specific binding of membrane proteins-lipids-metal ions in the plants’ responses to HMS. Lipids as metal ion recognizers first sense metal ion stimulation, and then ion channels and transporters specifically bind to those metal ions to control their movements and activities. This may provide new insights into the mechanisms of plant metal homeostasis.

Meanwhile, there are hundreds of phospholipids in the cells. They vary in the head groups (charge, polarity, shape, size), length and saturation aspects of the fatty acid tails that determine the lipid bilayer hydrophobic cores. The cell membrane is also rich in non-phospholipids and fatty acids, among which cholesterol alone accounts for about 20% of the total lipid contents. These complex membrane compositions further produce complex lipid properties, such as membrane curvature, bilayer asymmetry, and hydrophobic mismatches. Most lipid properties are related to protein activities. In particular, it has become increasingly recognized over the last few decades that lipids can bind specifically to the membrane proteins like transporters to influence protein functions in addition to changing the physical properties of the lipid bilayers (such as fluidity, tension, hydrophobic defects, etc.). Thus, research challenges remain in understanding the biophysical and molecular mechanisms behind the regulation of membrane proteins by lipids. Therefore, the mechanisms underlying the membrane protein-lipid-metal ion-specific binding and its role in plant metal homeostasis needs to be further studied. In the future, we can further study the heterogeneity of lipid composition and the influence of lipid molecules on the structure, dynamics, and functions of membrane proteins in relation to plants’ tolerance to HMS.

## Figures and Tables

**Figure 1 membranes-11-00984-f001:**
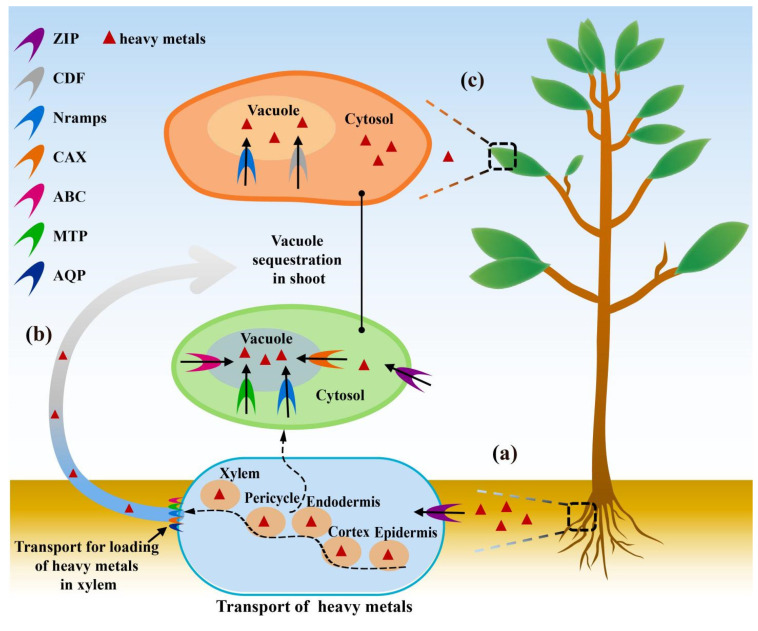
The role of transporters in plant responses to HMS. (**a**) Absorption of HMs in the roots by Zrt/Irt-like protein (ZIP). (**b**) Transport and loading of HMs from root to shoot, and their sequestration in vacuole as accomplished by various transporters such as ATP-binding cassette (ABC), cation/proton exchanger (CAX), natural resistance-associated macrophage protein (NRAMP), metal tolerance protein (MTP), and aquaporin (AQP). (**c**) Vacuolar sequestration of HMs in leaves by cation/proton exchanger (CDF) transporters.

**Figure 2 membranes-11-00984-f002:**
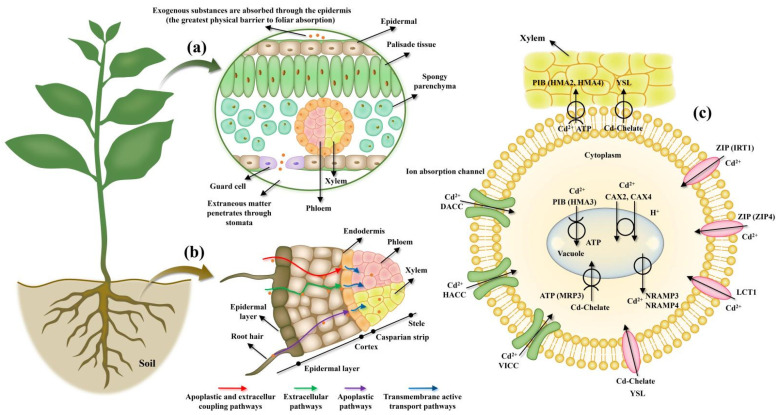
Absorption and transport of HMs by leaves and roots. (**a**) The absorption process of HMs by plant leaves. (**b**) The transport process of HMs absorbed by plant roots to the aboveground issues. (**c**) Specific proteins that control Cd’s ability to enter the root symplast, Cd’s isolation in vacuoles and its transport to the xylem [50]. DACC: depolarization-activated calcium channels; HACC: hyperpolarization-activated calcium channels; VICC: voltage-insensitive cation channels.

**Figure 3 membranes-11-00984-f003:**
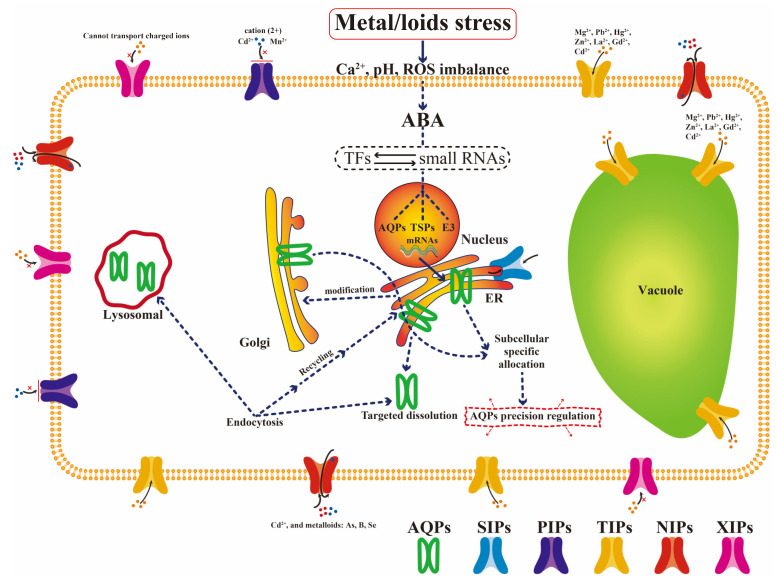
Coordinated response regulation model of AQPs of different subgroups under HMS. The underlying regulatory mechanisms are simply explained, including transcription, translation, inhibition, degradation, endocytosis, and exocytosis. SIPs: subcellular localization in the plasma membrane and ER membrane [71]. PIPs: plasma membrane; prevent the influx of divalent metal ions into the cytoplasm, especially Cd and Mn [72]. TIPs: vacuolar membrane; plasma membrane; represents the TIPs: vacuolar membrane; plasma membrane; represents the detoxification of compartmentalized HMs [73]. NIPs: plasma membrane; specific recognition and transport of As, B, Se and other metals, such as Cd ions [74]. XIPs: plasma membrane; charged ions may not be transported [75]. TSPs: tryptophan-rich sensory proteins; E3: Rma1H1 (a RING membrane anchor E3 ubiquitin ligase).

**Figure 4 membranes-11-00984-f004:**
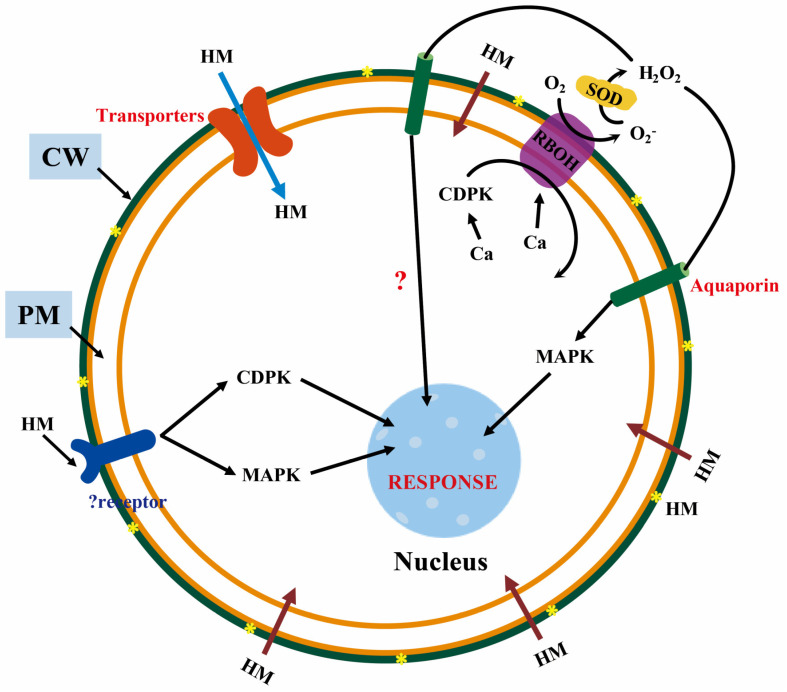
The signal cascade of the plant response to HMS. HMs enter the apoplast (brown arrow) through passive diffusion. The apoplast HMs transports proteins into the symplast (blue arrow) or stimulates unknown receptors to activate the MAPK and CDPK signals, thus causing nuclear transcription factor activation under HMS. In addition, the Ca activates CDPK and induces phosphorylated NADPH oxidase (RBOH) to produce O_2_, which is converted to H_2_O_2_ under the action of superoxide dismutase (SOD). The H_2_O_2_ enters the symbiote through aquaporins and then causes nuclear transcription factor activation in two ways. One way is unknown, and the other is to activate the MAPK pathway. The yellow star, “*”, stands for HMs interacting with cell wall polysaccharides in the CW.

**Figure 5 membranes-11-00984-f005:**
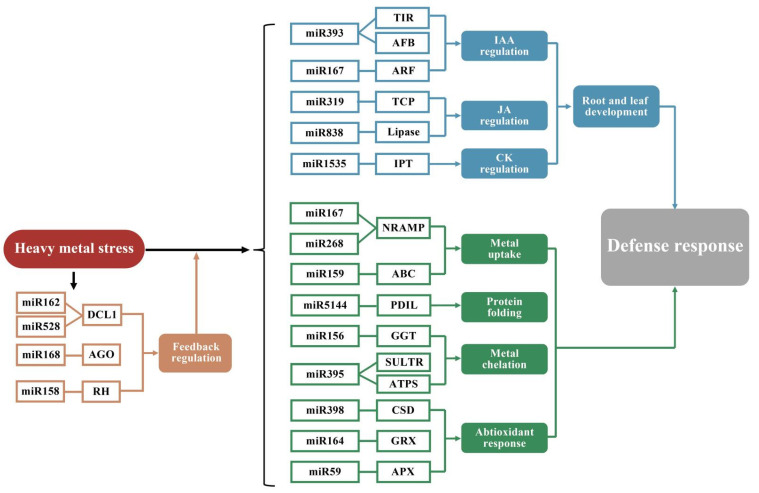
Regulatory networks of miRNA under HMS in plants. Feedback regulation of miRNAs under HMS in plants is indicated by orange letters and lines. Blue represents the regulation of hormone signaling that inhibits the growth of roots and leaves under HMS; green refers to the metabolic pathways involved in miRNA, including the uptake, transport and chelation of HMs and protein folding and antioxidation.

**Figure 6 membranes-11-00984-f006:**
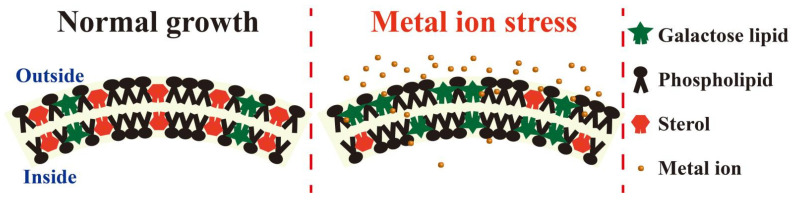
Changes in membrane fluidity under HMS. Plants change the adsorption of HM ions by changing the composition of membrane lipids, particularly by reducing the ratio of phospholipid sterol and contents of stigmasterol while increasing the proportion of galactose lipids. PL, phospholipid; GL, galactolipid.

**Figure 7 membranes-11-00984-f007:**
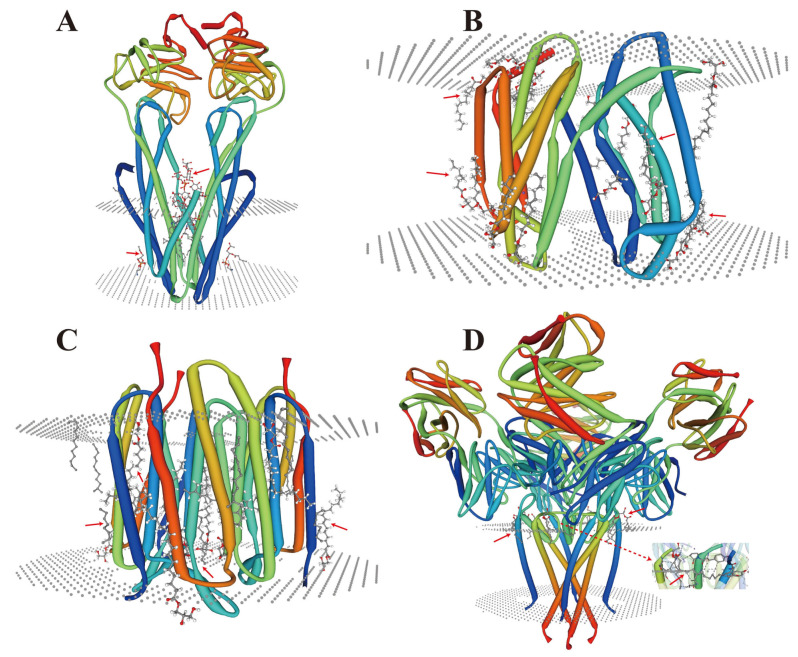
Crystal structure analysis of lipid-membrane protein interactions. (**A**) *Escherichia coli* (*E. coli*) MsbA in complex with LPS and inhibitor G907 (PDB ID: 4KSC) [147]; (**B**) Lipid II flippase MurJ, outward-facing conformation in *Thermosipho africanus* (PDB ID: 6NC9) [148]; (**C**) Steady-state-SMX dark state structure of the bacteriorhodopsin in *Halobacterium salinarum* (PDB ID: 6RQP) [149]; (**D**) Structure of the KcsA-G77A mutant or the 2,4-ion bound configuration of a K(+) channel selectivity filter in *Streptomyces lividans* (PDB ID: 6NFU) [150]. PDB, Protein Data Bank; MsbA, an essential ATP-binding cassette transporter in Gram-negative bacteria; LPS, lipopolysaccharide; MurJ, the lipid II flippase in *E. coli*. The red arrows indicate the positions where lipids and membrane proteins interact with each other.

**Figure 8 membranes-11-00984-f008:**
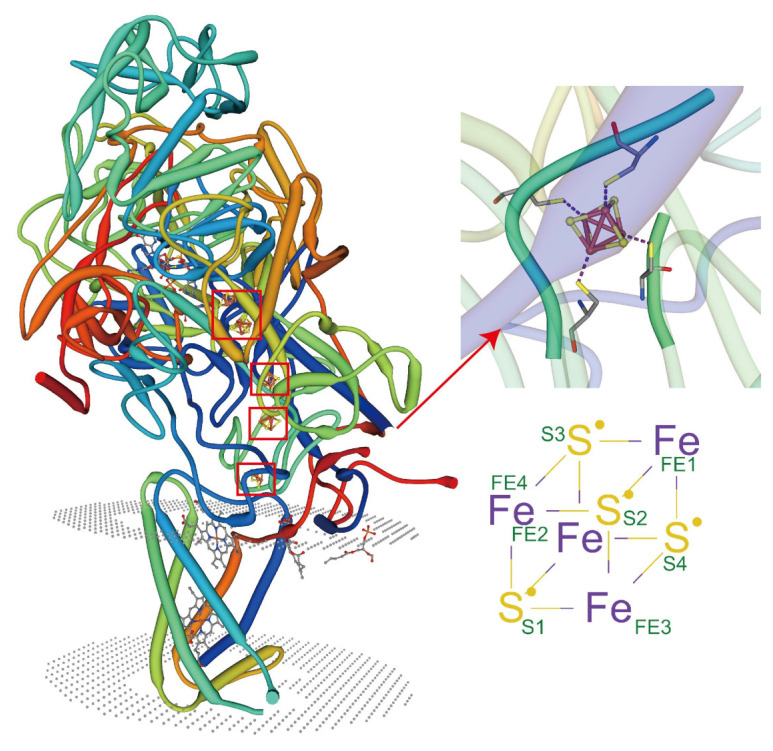
Lipids regulate the coordination of metal ions with membrane proteins. PDB ID: 3IR5, X-ray diffraction, 2.30 Å. Crystal structure of NarGHI mutant NarG-H49C in *Escherichia coli*. Tightly bound lipids exist between the transmembrane helices, while head groups bind in pockets composed of three subunits, and many positively charged residues participate in the binding.

**Figure 9 membranes-11-00984-f009:**
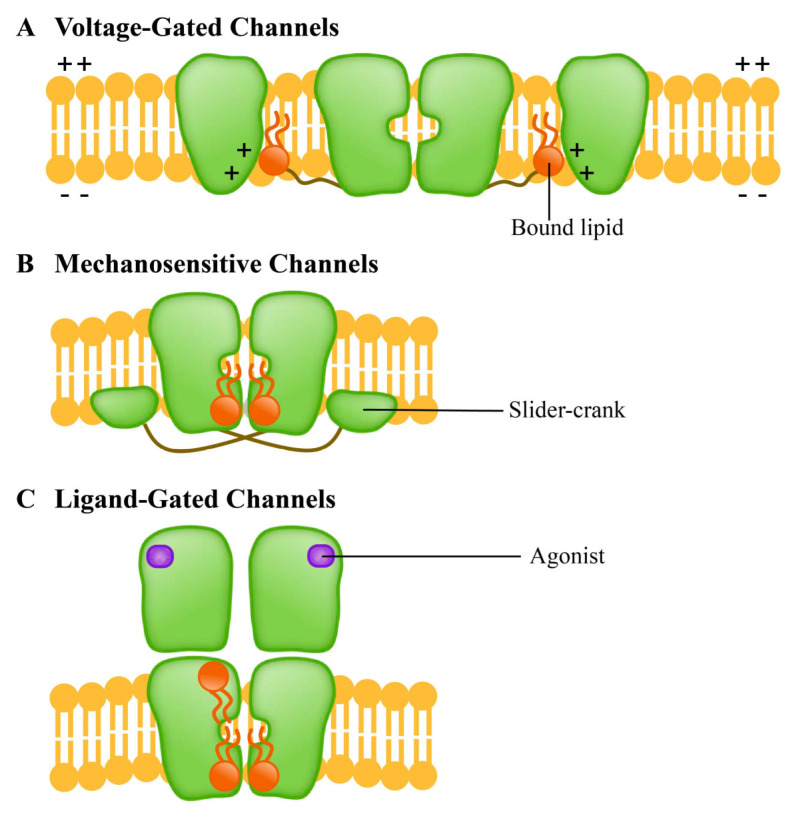
Different ion channel protein families have different lipid response patterns. (**A**) Voltage-gated channels (voltage-gated channel) is a tetramer; (**B**) Mechanosensitive channels (mechanically sensitive channel) uses the “crank slider” mechanism to convert increased double-layer tension into channel gating; (**C**) Ligand-gated channels (ligand gated channel) affects the function of TM helical stacking to change the channel function.

**Figure 10 membranes-11-00984-f010:**
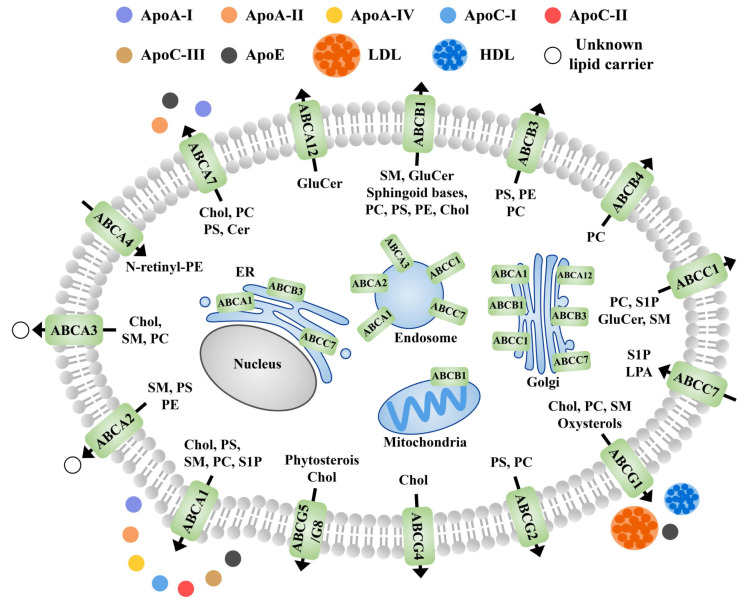
Schematic diagram of the subcellular localization, known receptors, and transport directions of ABC transporter proteins involved in lipid efflux in animals. Black arrows indicate the direction of transport across the plasma membrane. Apo, apolipoprotein; PS, phosphatidylserine; PC, phosphatidylcholine; PE, phosphatidylethanolamine; Chol, cholesterol; HDL, high density lipoproteins; LDL, low density lipoproteins; GluCer, glucosylceramide; Cer, ceramide; SM, sphingomyelin; N-rentinyl phosphatidylethanolamine; LPA, lysophosphatidic acid; S1P, sphingosine-1-phosphate; N- retinyl-PE.

**Figure 11 membranes-11-00984-f011:**
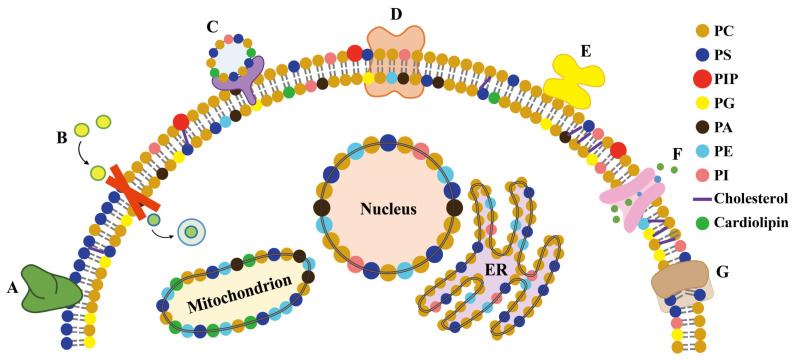
Proteins bind to lipids in different ways. (A) The peripheral protein binds to the hydrophobically anchored PS suppository with lipid specificity. (B) The integrin receptors involved in signal perception and transmission across cell membranes. (C) Proteins that induce vesicle fusion. (D) Integrins with local curvature induced by hydrophobic mismatch. (E) The peripheral proteins are attached to the membrane lipids, and their globular domains interact with the membrane interface regions without being embedded in the membrane. (F) The ion channels interact with cholesterol to control ion transport across the membrane. (G) The phospholipid-scrambling enzymes transport lipids through the membrane.

**Table 1 membranes-11-00984-t001:** Different members of the membrane protein family in different species bind specifically to metal ions (or complexes).

Membrane Protein Families	Proteins	Species	Metal Ion (Metal Complex)	References
ABCs	OsABCC1	*Oryza sativa*	As	[182]
	OsABCG36	*Oryza sativa*	Cd	[183]
	PtoABCG36	*Oryza sativa*	Cd	[184]
	AtABCC3	*Arabidopsis thaliana*	Cd	[185]
	AtATM3	*Arabidopsis thaliana*	Cd, Pb	[186]
	AtPDR8	*Arabidopsis thaliana*	Cd, Pb	[187]
	ARG1	*Oryza sativa*	Co, Ni	[188]
HMAs	BjHMA4	*Brassica juncea*	Cd	[189]
	OsHMA5	*Oryza sativa*	Cu	[190]
	OsHMA3	*Oryza sativa*	Zn	[191]
	PtoHMA5	*Populus tomentosa*	Cd	[192]
	CsHMA3/4	*Cucumis sativus*	Cd, Pb, Zn	[193]
	GmHMA8	*Glycine max*	Cu	[194]
	GmHMA3	*Glycine max*	Cd	[195]
	HvHMA1	*Hordeum vulgare*	Zn, Cu	[196]
NRAMPs	AtNRAMP6/3	*Arabidopsis thaliana*	Fe	[197,198]
	TtNRAMP6	*Triticum turgidum*	Cd	[199]
	SbNrat1	*Sorghum bicolor*	Al	[200]
	CjNRAMP1	*Crotalaria juncea*	Fe, Cd	[201]
	OsNRAMP1	*Oryza sativa*	As	[202]
	HvNramp5	*Hordeum vulgare*	Mn, Cd	[203]
	AhNRAMP1	*Arachis hypogaea*	Mn, Zn	[50]
ZIPs	OsZIP1	*Oryza sativa*	Zn, Cu, Cd	[204]
	OsZIP9/3/8	*Oryza sativa*	Zn	[205,206,207]
	NtZIP4A/B	*Nicotiana tabacum*	Zn, Cd	[208]
	ZmZIP5	*Zea mays*	Zn, Fe	[209]
	VsRIT1	*Vicia* sativa	Cd	[44]
MTPs	PtrMTP6	*Populus trichocarpa*	Mn, Co	[210]
	AtMTP11	*Arabidopsis thaliana*	Mn	[211]
	BnMTP3	*Brassica napus*	Mn, Zn	[212]
	CsMTP8.2	*Camellia sinensis*	Mn	[213]
	CsMTP7	*Camellia sinensi*	Fe	[214]
	CsMTP9	*Camellia sinensi*	Mn, Cd	[215]
	OsMTP11	*Oryza sativa*	Mn	[216]
OPTs	AtOPT6	*Arabidopsis thaliana*	Cd-GSH	[217]
	TcOPT3	*Thlaspi caerulescens*	Fe, Zn, Cd, Cu	[218]
	TcYSL3	*Thlaspi caerulescens*	Fe, Ni-NA	[219]
	SnYSL3	*Solanum nigrum*	Fe, Cu, Zn, Cd-NA	[220]
	AhYSL3.1	*Arachis hypogaea*	Cu	[221]
	HvYSL5	*Hordeum vulgare*	Fe-MA	[222]
	OsYSL2	*Oryza sativa*	Fe-NA, Mn-NA	[223]
	OsYSL6	*Oryza sativa*	Mn-NA	[224]
	OsYSL16	*Oryza sativa*	Cu-NA	[225]

**Table 2 membranes-11-00984-t002:** The metalloprotein databases and tools for calculating metal binding sites.

Databases and Tools	Description	Website	References
Metalloprotein databases			
InterMetalDB	Database and browser of intermolecular metal binding sites in macromolecules with structural information	intermetaldb.biotech.uni.wroc.pl/ (accessed on 12 November 2021)	[230]
MetalPDB	Database of metal sites in biological macromolecular structures	metalpdb.cerm.unifi.it/ (accessed on 12 November 2021)	[231]
PyCDB	Resource for phytochelatin complexes of nutritional and environmental metals	kuppal.shinyapps.io/pycdb (accessed on 12 November 2021)	[232]
ZifBASE	Database of zinc finger proteins and associated resources	web.iitd.ac.in/~sundar/zifbase (accessed on 12 November 2021)	[233]
MeLAD	Integrated resource for metalloenzyme-ligand associations	melad.ddtmlab.org/ (accessed on 12 November 2021)	[234]
ZincBind	Database of zinc binding sites, automatically generated from the Protein Data Bank.	zincbind.net/ (accessed on 12 November 2021)	[235]
CheckMyMetal	Database of metal binding site validation server	https://cmm.minorlab.org/ (accessed on 12 November 2021)	[236]
MACiE	Exploring the diversity of biochemical reactions	www.ebi.ac.uk/thornton-srv/databases/MACiE/ (accessed on 12 November 2021)	[237]
dbTEU	DataBase of Trace Element Utilization	gladyshevlab.bwh.harvard.edu/trace_element (accessed on 12 November 2021)	[238]
Computational tools			
MetalPredator	Fe/(Fe-S)	metalweb.cerm.unifi.it/tools/metalpredator/ (accessed on 12 November 2021)	[239]
RDGB	Fe, Cu, Zn and other metals	www.cerm.unifi.it/ (accessed on 12 November 2021)	[240]
ZINCCLUSTER	Zn	www.metalactive.in/ (accessed on 12 November 2021)	[241]
MPLs-Pred	Metal ions	icdtools.nenu.edu.cn/mpls_pred (accessed on 12 November 2021)	[242]
BioMetAll	identification of conformational changes that alter the formation of metal-binding sites; Metalloenzyme design	github.com/insilichem/biometall (accessed on 12 November 2021)	[243]
SECISearch3 and Seblastian	Se	seblastian.crg.eu/ (accessed on 12 November 2021)	[244]
SeqCHED	Mg, Co, Ni, Ca	oca.weizmann.ac.il/oca-bin/ocamain (accessed on 12 November 2021)	[245]
MIonSite	Mg, Mn, Fe, Cu, Co, Cd, Ni	github.com/LiangQiaoGu/MIonSite.git (accessed on 12 November 2021)	[246]
MIB	Mn, Co, Zn, Ni, Hg, Cd	bioinfo.cmu.edu.tw/MIB/ (accessed on 12 November 2021)	[247]

## Data Availability

Not applicable.

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
