# Peer review of "The Mechanism of Metal Homeostasis in Plants: A New View on the Synergistic Regulation Pathway of Membrane Proteins, Lipids and Metal Ions"

_membranes, 2021, doi:10.3390/membranes11120984_

Round 1

Reviewer 1 Report

The manuscript entitled “The mechanism of metal homeostasis in plants: a new view on the synergistic regulation pathway of membrane proteins, lipids and metal ions” is well organized and informative. However, a few minor corrections are required before publishing in a reputed journal.

  1. Line no. 28. Replace “will also harm” by “negatively effect”
  2. Line no. 29. delete “Most”
  3. The authors are suggested to change figure 1 and figure 2 with an improved version of the figures.
  4. The authors have not mentioned the NIP transporters in the whole of the manuscript. NIP transports also play a crucial role in metalloid uptake. Authors are suggested to incorporate the details of the NIP transporters and also incorporate NIP into figure 2.

Author Response

Thanks for the help and guidance of reviewers. We have revised the manuscript one by one according to your suggestions. Please refer to the attachment for details (Reviewr 1.pdf).

Reviewer 2 Report

The aim of this review (Manuscript ID: membranes-1462047) is to focus the mechanisms that regulate metal homeostasis in plant, pointing out a synergistic cooperation among membrane proteins, lipids and metal ions. The topic of the review is of considerable interest for the scientific community, however it has been developed with some kind of superficiality in most part of it. In general the review remains very much speculative without approaching the numerous problematic related to the uptake-radial movement of metal ions in roots “considering all cellular compartments” and the ion movement in xylematic flow for the active translocation to the aerial part of the plants. Recently the reorganization of plant endomembranes induces by heavy metals has been discussed in a review (https://doi.org/10.3390/plants9040482) and could help to frame the topic in a broader vision. Then, in the text, I suggested to introduce the aquaporins (AQPs) because they have not only a role in water transport and stress responses but also in maintaining lipid membrane architecture (https://doi.org/10.3389/fpls.2016.01322; https://www.frontiersin.org/articles/10.3389/fcell.2021.656438/full). AQPs are essential actor cell homeostasis and should be mentioned. I find interesting the report that NIP1.1 T-DNA mutant confers Arsenic tolerance not by blocking uptake but likely modifying the metalloid intracellular compartmentalization (https://doi.org/10.3389/fpls.2018.01949). Furthermore a large number of researches of Michael R. Blatt’s Laboratory (https://www.gla.ac.uk/researchinstitutes/biology/staff/michaelblatt/) may help to deepen the topic of the review.

In general, the review presents several inaccuracies (specified below) that prevent the reader from fully understanding of the manuscript topic. I am concerned about the complexity of the system so that I auspicate more clarity, where possible, in the present review. Some reported reference are unappropriated because referred to animal systems. For example, there are numerous studies on the chemical, biological, biochemical and molecular composition and dynamicity of membrane and in particular plasma membrane in plants. I don’t understand why the authors refer mainly to animal systems. This review is on metal homeostasis in plants.

The review requires an accurate English revision to improve the grammar and the clarity of some sentences. It is necessary more care in preparing the Tables, Figures and References, several references are reported twice in the Reference section.

Some comments are reported below:

Line 15 membrane proteins are linked to the highly complex… should be… membrane proteins interact with the highly complex

Line 71 Cell membrane is the barrier that prevents extracellular substances from entering the cell freely, and its main components are proteins and lipids. It is a very generic sentence. Better add: “and move among compartments”… Cell membrane is the barrier that prevents extracellular substances from entering the cell and move among compartments freely, and its main components are proteins and lipids.

Lines 72-76 Please rephrase the two sentences with the appropriate references.  

Line 87 What do you mean for apple in Figure 1? And where is Table 2? What is the significance of HM*? Figure 1 isn’t clear, what is the space between CW and PM?

Line 104 memb”ers… should be members

Line 106 Show… show

Figure 2 There is not the II in the Figure 2. What about vacuole sequestration in root? There is no reference in the figure 2!

Line 117 what do you mean for vacuum sequestration?

Line 119 The title is not clear

Line 120 rna… should be… RNA

Line 121 Please take off (delete) By binding to

Line 127 abtioxidant should be antioxidant

Line 134 It is better use uptake instead of absorption, then what are the plant metals?

Line 135 mirnas… should be… miRNA

Line 139 ATP thickness (APS)??? SULTR2:1 please explicit the acronym, for all the acronyms present in the manuscript.

Lines 139-140 they can produce GSH and PC genes… I suggest… they can induce the expression of GSH and PC genes.

Line 161 and line 170 what do you mean for promote the production of apoplastic barrier?

Line 168 Figure 4 V-AYPase… should be… V-ATPase.

Line 169 Crown?

Line 195 The use of the term “biofilm” is inappropriate and should be changed with “membranes”.

Line 220 Lipid changes are… lipid changes depend on

Lines 232-233 check the sentence, the meaning is not clear.

Line 254. Figure 6. It is important in the figure to highlight that the membranes of the nucleus and the ER are formed by a lipid bilayer like that of the plasma membrane. Furthermore, the mitochondrion membranes are formed by two lipid double layers.

Line 261-263 the sentence should be: Plants are inevitably exposed to environmental stresses such as metal contamination, and these elements may penetrate passively through aquaporins or they can promotes mechanisms to effectively perceive, emergency response response and stress adaptation mechanisms.

Line 267: take off “that” please

Line 295 plasma????

Lines 346-347 transport, transportation what do you mean?

Line 483 References n. 30 and n. 31 are the same references, the same happened also for 56 and 57. Reference n 84 is the same of reference n 108. There are a lot of reference referring to animal cells.

Author Response

Thanks for the help and guidance of reviewers. We have revised the manuscript one by one according to your suggestions. Please refer to the attachment for details (Reviewr 2.pdf).

Round 2

Reviewer 2 Report

To the Authors,

I have just received your new version of resubmitting manuscript in which following most of my suggestions you have corrected, modified and amplified it. However, I am sorry to inform you that this new version still requires diversified corrections, improvement of some sentences, a clarity of the figures. Probably, some figures can be omitted. Most of my comments are reported in the attached text below and I also recommend an accurate English revision made by an expert of the field. 

Author Response

Thanks to reviewer 2# for your guidance and help. We have uploaded the revised and improved manuscript, with a one-by-one reply for each suggestion, as shown in attachment Review 2.pdf.

Round 3

Reviewer 2 Report

I endorse the publication of the manuscript in the present form